# Consumption of Alcoholic Beverages and the Prevalence of Metabolic Syndrome and Its Components

**DOI:** 10.3390/nu11112764

**Published:** 2019-11-14

**Authors:** Edyta Suliga, Dorota Kozieł, Elzbieta Ciesla, Dorota Rebak, Martyna Głuszek-Osuch, Stanisław Głuszek

**Affiliations:** 1The Institute of Health Sciences, Medical College, Jan Kochanowski University (JKU), ul. Zeromskiego 5, 25-369 Kielce, Poland; dorota.koziel@ujk.edu.pl (D.K.); eciesla@ujk.edu.pl (E.C.); dorota.rebak@ujk.edu.pl (D.R.); mgluszekosuch@ujk.edu.pl (M.G.-O.); 2The Institute of Medical Sciences, Medical College, JKU, ul. Zeromskiego 5, 25-369 Kielce, Poland; sgluszek@wp.pl

**Keywords:** alcohol consumption, metabolic syndrome, men, women

## Abstract

The relationship between alcohol consumption and the prevalence of metabolic syndrome is not consistent and may vary between populations, depending on age, sex, ethnicity, cultural traditions and lifestyle. We have hypothesized that moderate alcohol consumption will be associated with the lowest risk of the syndrome. The aim of the present study is to examine the relationship between the current consumption of alcohol and the prevalence of metabolic syndrome and its components. The research material includes data obtained from 12,285 men and women, in the age range of 37–66 years. Multiple logistic regression was used in the statistical analysis. Metabolic syndrome (MetS) was defined according to the International Diabetes Federation. In men, a current consumption of >30 g of alcohol/day was significantly associated with a higher risk of metabolic syndrome (OR = 1.73, 95% CI = 1.25–2.39), high blood pressure (OR = 2.76, 95% CI = 1.64–4.65), elevated glucose concentration (OR = 1.70, 95% CI = 1.24–2.32), and abdominal obesity (OR = 1.77; 95% CI = 1.07–2.92). In women, the consumption from 10.1 to 15.0 g of alcohol was associated only with a higher risk of abnormal glucose concentration (OR = 1.65; 95% CI = 1.14–2.38.) In both sexes, current alcohol consumption was associated with higher high-density lipoproteins (HDL)-cholesterol concentration (*p* < 0.05). No relationship was found between alcohol consumption and triglyceride concentration. It is difficult to formulate unequivocal recommendations regarding alcohol intake in MetS prophylaxis due to its different association with particular MetS components. In order to explain the causal relationship between alcohol consumption and MetS and its components, prospective studies are necessary.

## 1. Introduction

Alcohol intake is one of the seven main risk factors in the world for both deaths and disability adjusted life years (DALYs) [1]. Its consumption is associated with more than 60 acute and chronic diseases. Some studies, however, point out the protective effects of alcohol in cardiovascular diseases [2,3,4], diabetes [5,6], and kidney cancer [7].

The association between drinking alcohol and the prevalence of metabolic syndrome (MetS) and its components is not consistent. One prospective study found a linear increase in metabolic syndrome risk with an increase in alcohol consumption [8]. Park showed that alcohol consumption is associated with a higher incidence of MetS in men. However, no such association was found in women [9]. In a longitudinal study of older people, alcohol consumption was not associated with the prevalence and incidence of MetS compared with abstainers in both sexes. In men, however, there was an adverse effect of alcohol on glycemia, waist circumference, and systolic blood pressure [10]. Alkerwi et al. found, however, that moderate alcohol intake reduced the prevalence of MetS, while in women the beneficial effect of alcohol was noted at a dose twice lower than that in men [11]. In a study conducted by Stoutenberg et al., alcohol reduced the risk of MetS in men at every level of intake [12]. There were also links between moderate and heavy alcohol consumption and a high concentration of fasting glucose and a reverse dose-response association between alcohol and a low concentration of high-density lipoproteins (HDL)-cholesterol. Alcohol intake was not significantly associated with abdominal obesity, hypertension, and elevated triglyceride concentration. Freiberg et al. showed that moderate alcohol consumption was associated with a lower incidence of MetS and a favorable effect on serum lipid concentration and waist circumference [13].

Patterns of alcohol consumption and their relationships with health may vary in different populations, depending on age, sex, ethnicity, cultural traditions, socioeconomic status and lifestyle [14,15,16]. The results of previous studies have shown that moderate alcohol consumption was in many populations associated with a lower prevalence of cardiovascular disease (CVD), and a lower mortality due to them [2,4]. Because the presence of MetS leads to an increased risk of CVD, we hypothesized that the association between alcohol consumption and the prevalence of MetS will be similar to that for CVD, i.e., moderate consumption will be associated with a lower odds ratio of MetS. Here, our aim was to examine the relationship between the current consumption of alcoholic beverages and the prevalence of MetS and its components in men and women from the Świętokrzyskie region in Poland.

## 2. Subjects and Methods

### 2.1. Study Design and Sample Collection

The research material that was analyzed was the data of 13,172 participants of the PONS project (Polish-Norwegian Study), which was carried out in the years 2010 to 2012 in the Świętokrzyskie region in Poland. Men and women aged 37–66 participated in this cross-sectional study regarding health. We have described in detail the recruitment process as well as the research method and techniques used in previous papers [17,18]. The project involved biochemical and anthropometric measurements. The sociodemographic data of the participants and information on health and lifestyle were collected in face-to-face interviews using structured questionnaires. Out of the 13,172 people who volunteered for the study, the data of 12,285 participants (66.7% women) were used for further analysis. The remaining 887 subjects were rejected due to a lack of complete data.

### 2.2. Ethical Approval

The Ethics Committee from the Cancer Centre and Institute of Oncology in Warsaw approved the study (No. 69/2009/1/2011). Permission to analyze the above data was issued in 2016 by the Committee on Bioethics at the Faculty of Health Sciences, Jan Kochanowski University, Kielce, Poland, (No. 45/2016). 

### 2.3. Biochemical Profiling and Anthropometric Measurements

All laboratory analyses of fasting plasma glucose, triglycerides (TGs) and high-density lipoprotein cholesterol (HDL-cholesterol) concentrations were performed in one hospital laboratory that meets the standards of the National Reference Laboratory. The glucose concentration in the blood serum was measured using the enzyme method with hexokinase. The concentration of TGs was assessed using the enzymatic method, with phosphoglycerol oxidase and determination of H_2_O_2_ (with peroxidase). The concentration of HDL-cholesterol was obtained using the colorimetric non-precipitation method with polyethylene glycol modified enzymes (PEG method, Kyowa, Medex). The laboratory tests were performed with the Integra 800 instrument (La Roche Diagnostics, Switzerland). The coefficient of variation (CV) for the parameters determined in the internal quality control were 1.7% for glucose, 2.7% for HDL-cholesterol, and 1.5% for TG. The variation coefficient of the CV (V2) corrected set in the external quality control for the methods used in the study was as follows: For glucose, 2.5%, for HDL-cholesterol, 4.6% and for TG, 2.5%. The internal quality control of the tested parameters was based on the following control materials: PreciControl ClinChem Multi 1, reference number: 05947626190; PreciControl ClinChem Multi 2, reference number: 05947774190. A blood pressure (BP) measurement on the right upper limb artery in a sitting position was performed twice. Here, anthropometric measurements included body height measurements, using a stadiometer (SECA GmbH & Co., Hamburg, Germany). To measure body weight, a body composition analyzer (Tanita S.C. 240MA, Tokyo, Japan) was used, and waist circumference was understood as half the distance between the lower edge of the rib and the upper iliac crest, which was measured with a non-elastic metric tape measure. The assumed measuring accuracy was 0.1 cm and 0.1 kg. Body height and weight measurements were used to calculate the body mass index (BMI, kg/m^2^).

### 2.4. MetS Definition

MetS was defined according to the International Diabetes Federation (IDF) Task Force for Epidemiology and Prevention guidelines. MetS classification, according to IDF, unlike other classifications, takes into account ethnic differences of waist circumferences. Due to the lower cut-off points defining abdominal obesity, this classification allows earlier diagnosis of MetS, i.e., with less excess weight. It also identifies individuals with MetS who have the correct BMI more accurately. MetS was diagnosed for subjects with at least three of five of the following components: Abdominal obesity (waist circumference ≥94 cm in men and ≥80 cm in women), elevated BP (systolic BP ≥130 and/or diastolic BP ≥85 mm, Hg or antihypertensive drug treatment, or previously diagnosed hypertension), elevated fasting glucose (≥100 mg/dL or drug treatment of elevated glucose or previously diagnosed type 2 diabetes), hypertriglyceridemia (≥150 mg/dL or drug treatment for elevated TG or previously diagnosed dyslipidemia), and reduced HDL-cholesterol (<40 mg/dL in men, <50 mg/dL in women, or drug treatment for reduced HDL-cholesterol or previously diagnosed dyslipidemia) [19]. The following drugs were included in the treatment of MetS components in the case of hypertension: Beta-blockers, angiotensin-converting enzyme inhibitors, angiotensin receptor blockers, and calcium channel blockers. In the case of type 2 diabetes, oral hypoglycemic agents, insulin, and incretin drugs were administered subcutaneously. In the case of dyslipidemia, statins and fibrates were used.

### 2.5. Alcohol Consumption

Alcohol intake was obtained using a standardized questionnaire that included questions on the participants usual consumption of vodka, beer, and wine during the previous year. For every participant, a daily intake of pure ethanol was calculated. The calculation took into account the average content of ethanol in each type of drink consumed and the frequency of its consumption. Since there are significant differences in the amount of alcohol consumed by men and women, different categories of consumption in both sexes were used in the analysis. All individuals were divided into five groups: Nondrinkers (participants who did not drink any alcohol during the previous 12 months) and drinkers, in bands of 0.1–10.0, 10.1–20.0, 20.1–30.0, and >30.0 g of ethanol/day in men, and 0.1–5.0, 5.1–10.0, 10.1–15.0, and >15 g of ethanol/day in women. 

### 2.6. Sociodemographic and Lifestyle Data

The sociodemographic variables included sex, age (years), education (total number of education years), and marital status (married or in a stable relationship, single, or a widow/widower). Subjects were classified as current, former, and non-smokers. Current smokers were respondents who smoked cigarettes every day during the study, former smokers were those who had not smoked for longer than six months. The rest of the participants comprised the group of non-smokers. As one serving of coffee, one standard 250 mL mug was adopted. Physical activity (PA) was evaluated with the use of the International Physical Activity Questionnaire (IPAQ). Here, the IPAQ was used to collect information on the frequency and duration of all physical activities (intensive, moderate, and walking) done by respondents during the last week, connected with their professional work, housework, active movement, recreation, and sports. Total PA was presented in metabolic equivalents (MET/min/week^−1^) [20].

### 2.7. Statistical Analyses

All continuous variables were expressed as means and standard deviations (X ± SD). The categorical variables were reported as frequencies and percentages (*n*, %). Differences in baseline characteristics between individuals with diagnosed MetS and the control group were evaluated through the Mann–Whitney U test or the chi-square test, depending on the distribution of each feature. The analyses were performed separately for both sexes. The multivariate logistic regression analyses were used to estimate the odds ratios (ORs) and 95% confidence intervals (CIs) for the prevalence of MetS. Non-drinkers were adopted as the reference group. Analyses were repeated with each MetS component in the binary form, based on standardized cutoff values (0 = normal, 1 = abnormal). Three models were adopted in the logistic regression analysis: Model I—unadjusted; model II—adjusted for age, education, physical activity, coffee consumption (continuous variables) and marital status (married or in a relationship; single or a widow/widower), smoking (current, past or never); and Model III—additionally adjusted for BMI. A *p* value <0.05 was assumed statistically significant for all calculations. All data were analyzed using the Statistical Package Statistica software (version 13.1, StatSoft, Warsaw, Poland).

## 3. Results

Men who were diagnosed with MetS were slightly older, more often married or living in stable relationships, had a heavier weight and higher BMI, more often had abdominal obesity, high BP, increased concentration of glucose and TG and a reduced HDL-cholesterol concentration, compared to men from the control group (Table 1). However, they did not differ significantly in their level of education. Men with MetS more often consumed larger amounts of alcohol (>30.0 g/day), more often were former smokers, but also more non-smokers, compared with men from the control group. Women who were diagnosed with MetS were older, characterized by lower education and were less likely to be married or living in a stable relationship, compared to those in the control group. Women with MetS had significantly greater mass and BMI, more often had abdominal obesity, high BP, increased concentration of glucose and TG, and decreased HDL-cholesterol concentration compared to women in the control group. In the group with MetS there were more non-drinkers, but no significant differences were noted in the percentage of current and former smokers. Both men and women with MetS drank less coffee and were less physically active compared to participants in the control group.

In the group of men currently consuming the largest amounts of alcohol, (>30 g/day) the risk of MetS was significantly higher compared to non-drinkers in all three models (Table 2). In women, with the increase of alcohol consumption MetS risk decreased, but only in the unadjusted model. The inclusion of confounding variables caused that the association was no longer statistically significant.

The highest alcohol consumption in men (>30 g/day) was associated with a greater risk of abdominal obesity, but only in model II, which adjusted for age, marital status, education, smoking, drinking coffee and physical activity (Table 3). The addition of BMI to the model meant that this relationship ceased to be statistically significant. In women, the risk of abdominal obesity was lower at every level of alcohol consumption, but only in the unadjusted model.

Consumption of more than 30 g of alcohol per day increased the risk of high blood pressure in men in all three models (Table 4). In women, drinking 0.1 to 10 g of alcohol per day was associated with a reduced risk of high blood pressure compared to non-drinkers in the unadjusted model. In both adjusted models there was no significant relationship between alcohol consumption and BP.

The highest alcohol consumption significantly increased the risk of elevated glucose concentration in all three models in men (Table 5). In women in the unadjusted model, consumption of more than 15 g of alcohol/day significantly reduced the risk of elevated glucose concentration, while consumption of 10.1–15.0 g increased the risk in both adjusted models.

In men who drink >10 g alcohol/day, the risk of reduced HDL-cholesterol concentration decreased significantly with the increase in intake in both the unadjusted model and the two adjusted models (Table 6). In women, in the unadjusted model, at every level of alcohol consumption, the risk of decreased HDL-cholesterol concentration was significantly lower than in non-drinkers. In model II, a similar relationship was found in the group of drinkers >5 g, and in the group of drinkers >15 g alcohol/day in model III.

There were no significant relationships between alcohol consumption and triglyceride concentration in men (Table 7). In women, the concentration of triglycerides was lower at every level of alcohol consumption in comparison to non-drinking only in the unadjusted model.

## 4. Discussion

The study shows the existence of links between the current alcohol consumption and MetS risk, but the strength and direction of these relationships may differ in both sexes and may be different for individual MetS components. In the analysis, we adopted a half-dose of alcohol for women compared to men due to differences in body composition and alcohol metabolism between men and women. Nevertheless, we did not find in women, similar to men, an increased risk of MetS and its components (except glucose) in the highest categories of alcohol consumption. This may be due to the fact that women drinking more than 10 g of ethanol/day were more than three times less prevalent than men drinking more than 20 g of ethanol/day respectively. The increased risk of MetS in the highest category of alcohol consumption in men, and the lack of such association in women, may also be the result of other behaviors in the women surveyed which are more favorable to health, e.g., a lower percentage of current and former smokers, as well as a healthier diet. The results of other authors are also quite diverse. In the Korean population, consumption of 0.1 to 5.0 g of alcohol/day was associated with a significantly lower prevalence of MetS [21]. However, Alkerwi et al., in the meta-analysis of observational studies, noted that an intake of <40 g alcohol/day in men and <20 g in women significantly reduces the risk of MetS [11]. Wakabayashi et al. found that the risk of MetS was the lowest in subjects drinking <22 g of alcohol/day in both sexes [22]. The results of our analyses indicate that similar associations were found only in women in the unadjusted model. In an earlier study conducted in Poland, it was also not noted that drinking alcohol was associated with a lower risk of MetS in men [23]. It was only observed that a lower probability of MetS appeared in women in the highest quartile of alcohol consumption. The authors mentioned, however, that the overall amount of alcohol consumed by women in this study was small, as with 90% of them, the consumption did not exceed 3 g of ethanol per day. The results of our research are, at least partially, in line with the results of a meta-analysis of prospective studies, which proved that heavy consumption of alcohol, i.e., >35 g/day, is associated with higher MetS risk [24]. Also, Bermudez et al. confirmed in a multivariate analysis that men consuming 28.41–47.33 g alcohol/day had an increased risk of MetS [25]. Barrio-Lopez et al. found an increased risk of MetS in those who drink ≥7 drinks/week, which was mainly due to beer consumption [26]. There was no significant link between drinking wine or liquor and MetS. Lee et al. have indicated, however, that the average frequency of drinking is not associated with the prevalence of MetS in any of the sexes, while a factor which positively correlated with some of the MetS components was the frequency of binge drinking [27].

The results of the research on the relationship between alcohol consumption and individual MetS components are also ambiguous. Freiberg et al. showed a lower risk of abdominal obesity in both drinkers of ≥1 and ≥20 alcoholic drinks per month, compared to drinkers of <1 alcoholic drink/month [13]. In turn, Lourenço et al. found an increased risk of abdominal obesity in women who drink 15.1–30 g/day and >30 g/day and in men who drink >60g/day, compared to non-drinkers [28]. This relationship appeared with a much higher alcohol consumption than in the population we studied. Similarly, the adverse effect of alcohol on the waist circumference was observed in older men (i.e., those aged 65–84 years), but in women no similar relationship was observed [10]. Alcohol consumption may probably contribute to excessive weight gains in some people as a result of increased energy supply from alcoholic beverages [29]. Schröder et al. have stated that after controlling for energy under-reporting, drinking more than 30 g of ethanol per day significantly increases the risk of exceeding the recommended energy intake for men [30]. Alcohol can also contribute to abdominal obesity through non-calorie-related mechanisms, such as changes in the concentration of steroid hormones that may cause central fat storage [31]. Ambiguous associations between alcohol consumption and abdominal obesity may also result from various effects caused by different alcoholic beverages. Vadstrup et al. found that moderate or high consumption of beer and spirits were associated with an increased waist circumference, while consumption of moderate-to-high amounts of wine had the opposite effect [32].

Yoon et al. observed that the daily intake of ≥30 g of alcohol increased blood pressure in men [33], which is consistent with the results of the studies in our population. Meta-analysis of trials conducted by Roerecke et al. confirmed that a reduction in alcohol consumption reduces BP in a dose-dependent manner [34]. For people who drank more than 2 alcoholic drinks a day, a decrease in alcohol consumption was associated with a significant reduction in BP. The authors emphasize that the above associations are less documented in women [34].

Stoutenberg et al. observed links between moderate and heavy alcohol consumption and a high concentration of fasting glucose in men [12], while in the Korean population it was found that heavy alcohol consumption was associated with a high glucose concentration in women [21]. Nygren et al. found that binge drinking and total alcohol consumption were associated with increased glucose concentration in women in Sweden [35]. On the other hand, in some studies, a beneficial effect of the intake of moderate amounts of alcohol on the risk of diabetes has been observed [5,6]. Huang et al. have associated the varied risk of diabetes in the studies of various authors with the type of alcoholic beverages they drink [36]. These authors have shown that drinking wine can have a much stronger protective effect on the risk of type 2 diabetes than beer or spirits. The increased risk of abnormal fasting glucose concentration in our studies may partly result from the fact that the pattern of consumption of alcoholic beverages in Poland is dominated by beer, which provided more than 50% of alcohol consumed in total [37]. In the second place there are spirits, providing over 30% of consumed alcohol. The lowest is consumption of wine, which is attributed the greatest health benefits among alcoholic beverages [38,39].

Current research suggests that the relationship between alcohol consumption and triglyceridemia is J-shaped, and triglyceridemia is lowest in people who drink 10–20 g/alcohol per day [40]. In the men we examined, there was a similar (but not significant) tendency in the highest analyzed category of alcohol consumption. In a study previously carried out in Poland, with the increase in alcohol consumption, a reduced likelihood of elevated triglycerides among women was observed [23]. However, the results of our research are consistent with the results of the meta-analysis carried out in 2011 [41]. Its authors also did not confirm the significant effect of alcohol on TG concentration. This may be due to the fact that the consumption of moderate amounts of alcohol may increase lipoprotein lipase activity, and thus influence triglyceridemia [40].

In a long-term 6-year study, a non-linear, umbrella-shaped relationship between alcohol consumption and HDL-cholesterol concentration was reported [42]. HDL-cholesterol concentration was the lowest in people who consumed moderate amounts of alcohol (1–2 servings/day in men and 0.5–1.0 servings/day in women). The umbrella-shaped relationship was the result of consumption of mainly hard liquor, and not of beer, where the relationship changes in HDL-cholesterol concentration were linear. In cross-sectional analysis, the authors observed a dose-dependent relationship response between higher alcohol consumption and higher baseline HDL-cholesterol levels, which is consistent with the results of our research. Similar results were obtained by most other authors [21,23,24,33]. Alcohol intake may increase the concentration of HDL-cholesterol, probably by increasing liver production and/or the transport rate of HDL apolipoproteins apoA-I and apoA-II, as well as by increasing cellular cholesterol outflow and cholesterol esterification in plasma [43,44,45].

The limitation of this study is, above all, the cross-sectional design of the study. There is also the risk of imprecisely determining the amount and frequency of alcoholic beverages drunk by the study participants. In addition, among non-drinkers, we were unable to separate recent abstainers from lifetime abstainers. Interpreting the results regarding the relationship between alcohol consumption and MetS risk may also be hindered by the multidirectional effects of alcohol on individual components whose MetS is a compilation. Sun et al. even suggest that it is more appropriate to estimate the impact of alcohol consumption on individual components than on MetS as a whole [24]. In addition, the obtained results may be influenced by the differences in the utilization of alcohol in the body related to sex [46,47], ethnicity [48,49,50,51], and different effects caused by particular types of alcoholic beverages dominating in the consumption structure of a given population [32,36,38,39]. The limitation of our work is also the lack of data on the energy value of diets and nutrient intake, which may affect the spread of MetS. The strength of the study is taking into account a large number of confounders associated with MetS and its components, and a large number of participants (over 12,000), who are homogeneous in terms of their age and ethnicity. 

## 5. Conclusions

In men, a current consumption >30 g of alcohol per day was significantly associated with a higher risk of MetS, high blood pressure, increased glucose concentration, and abdominal obesity. In women, alcohol consumption from 10.1 to 15.0 g was only associated with a greater risk of abnormal glucose concentration. In both sexes, current alcohol consumption was associated with a higher concentration of HDL-cholesterol, while no significant associations between alcohol intake and triglyceride concentration were found.

It is difficult to formulate unequivocal recommendations regarding alcohol consumption in MetS prophylaxis due to its different impact on particular MetS components. In order to explain the causal relationship between alcohol consumption and the metabolic syndrome and its components, prospective studies are necessary.

## Figures and Tables

**Table 1 nutrients-11-02764-t001:** Sociodemographic characteristics and lifestyle habits of men and women.

Parameters		Men			Women		
		Control Group (*n* = 1933)	MetS Group (*n* = 2153)	*p*	Control Group (*n* = 4700)	MetS Group (*n* = 3499)	*p*
**Age (years) (X ± SD)**		55.14 ± 5.50	56.68 (5.24)	**<0.001** ^a^	54.33 ± 5.33	57.18 ± 4.92	**<0.001** ^a^
Education (years) (X ± SD)		13.30 ± 3.22	13.13 ± 3.17	**0.084** ^a^	13.67 ± 3.14	12.66 ± 3.12	**<0.001** ^a^
Body mass (kg) (X ± SD)		80.06 ± 11.56	90.35 ± 12.31	**<0.001** ^a^	67.55 ± 11.23	76.81 ± 12.95	**<0.001** ^a^
Body mass index (BMI) (kg/m^2^) (X ± SD)		26.71 ± 3.45	30.04 ± 3.68	**<0.001** ^a^	26.28 ± 4.56	30.27 ± 4.92	**<0.001** ^a^
Married or in a relationship: *n* (%) Single or widow/widower: *n* (%)		1688 (87.33) 245 (12.67)	1952 (90.66) 201 (9.34)	**0.001** ^b^	3538 (75.28) 1162 (24.72)	2564 (73.28) 935 (26.72)	**0.040** ^b^
Abdominal obesity: *n* (%)		922 (47.70)	1983 (92.10)	**<0.001** ^b^	2828 (60.17)	3344 (95.57)	**<0.001** ^b^
Elevated blood pressure or antihypertensive drug treatment: *n* (%)		1318 (68.18)	2049 (95.17)	**<0.001** ^b^	2558 (54.43)	3181 (90.91)	**<0.001** ^b^
Increased glucose concentration or drug treatment of elevated glucose: *n* (%)		347 (17.95)	1509 (70.09)	**<0.001** ^b^	345 (7.34)	1898 (54.24)	**<0.001** ^b^
Decreased HDL-cholesterol concentration or drug treatment for reduced HDL-cholesterol: *n* (%)		63 (3.26)	989 (45.94)	**<0.001** ^b^	296 (6.30)	2254 (64.42)	**<0.001** ^b^
Increased triglycerides concentration or drug treatment for elevated triglycerides: *n* (%)		195 (10.09)	1445 (67.12)	**<0.001** ^b^	243 (5.17)	2331 (66.62)	**<0.001** ^b^
Alcohol consumption (g/day): *n* (%)							
Nondrinkers (Men)	Nondrinkers (Women)	265 (13.71)	289 (13.42)	**0.015** ^b^	746 (15.87)	693 (19.81)	**<0.001** ^b^
0.1–10.0	0.1–5.0	1070 (55.35)	1165 (54.11)		3328 (70.81)	2432 (69.51)	
10.1–20.0	5.1–10.0	358 (18.52)	399 (18.53)		413 (8.85)	267 (7.63)	
20.1–30.0	10.1–15.0	154 (7.97)	151 (7.01)		98 (2.09)	55 (1.57)	
>30.0	>15.0	86 (4.45)	149 (6.92)		112 (2.38)	52 (1.49)	
Current smokers: *n* (%)		444 (22.97)	436 (20.25)	**<0.001** ^b^	858 (18.26)	646 (18.46)	**0.734** ^b^
No smokers: *n* (%)		789 (22.97)	662 (30.75)		2489 (52.96)	1823 (52.10)	
Former smokers: *n* (%)		700 (36.21)	1055 (49.00)		1353 (28.78)	1030 (29.44)	
Coffee consumption (portion/day) (X ± SD)		1.88 ± 1.94	1.62 ± 1.85	**<0.001** ^a^	2.05 ± 1.86	1.68 ± 1.72	**<0.001** ^a^
Physical activity (METs/min/week^−1^) (X ± SD)		4984.0 ± 4154.9	4319.6 ± 3742.7	**<0.001** ^a^	4600.6 ± 3532.2	4204.0 ± 3381.3	**<0.001** ^a^

*n*—number of participants; X ± SD—arithmetic mean ± standard deviation; the numbers in bold indicate statistically significant results; ^a^ Mann–Whitney U test; ^b^ chi-square test. HDL: High-density lipoproteins. MetS: Metabolic syndrome. MET: Metabolic equivalent.

**Table 2 nutrients-11-02764-t002:** Odds ratios (OR) and 95% confidence intervals (CI) for MetS.

MetS Alcohol Consumption	Model I Unadjusted OR (95% CI)	*p*	Model II Adjusted OR (95% CI)	*p*	Model III Adjusted + BMI OR (95% CI)	*p*
Men						
Nondrinkers (ref.)	1.00		1.00		1.00	
0.1–10.0 g	0.99 (0.83–1.20)	0.986	0.99 (0.82–120)	0.920	0.99 (0.81–1.23)	0.973
10.1–20.0 g	1.02 (0.82–1.27)	0.846	1.05 (0.84–1.32)	0.646	1.01 (0.78–1.29)	0.963
20.1–30.0 g	0.90 (0.68–1.19)	0.456	0.98 (0.74–1.31)	0.896	0.94 (0.68–1.29)	0.697
>30.0 g of alcohol/day	**1.59 (1.16–2.17)**	**0.004**	**1.73 (1.25–2.39)**	**0.001**	**1.54 (1.08–2.19)**	**0.016**
Women						
Nondrinkers (ref.)	1.00		1.00		1.00	
0.1–5.0 g	**0.79 (0.70–0.88)**	**<0.001**	0.98 (0.87–1.11)	0.796	0.98 (0.86–1.12)	0.772
5.1–10.0 g	**0.69 (0.57–0.83)**	**<0.001**	1.02 (0.84–1.25)	0.788	1.11 (0.90–1.37)	0.345
10.1–15.0 g	**0.60 (0.43–0.85)**	**0.004**	0.92 (0.68–1.41)	0.920	1.12 (0.76–1.65)	0.579
>15.0 g of alcohol/day	**0.50 (0.35–0.71)**	**<0.001**	0.74 (0.52–1.06)	0.100	0.72 (0.49–1.06)	0.100

Model I—unadjusted; model II—adjusted for age, marital status, education, smoking, coffee consumption and physical activity; model III—adjusted for age, marital status, education, smoking, coffee consumption, physical activity and BMI. The numbers in bold indicate statistically significant results. Ref.: Reference level.

**Table 3 nutrients-11-02764-t003:** Odds ratios and 95% confidence intervals for abdominal obesity.

Abdominal Obesity Alcohol Consumption	Model I Unadjusted OR (95% CI)	*p*	Model II Adjusted OR (95% CI)	*p*	Model III Adjusted + BMI OR (95% CI)	*p*
Men						
Nondrinkers (ref.)	1.00		1.00		1.00	
0.1–10.0 g	0.98 (0.80–1.20)	0.829	0.98 (0.79–1.21)	0.841	1.06 (0.79–1.43)	0.698
10.1–20.0 g	1.04 (0.81–1.32)	0.780	1.07 (0.84–1.37)	0.590	0.95 (0.67–1.35)	0.790
20.1–30.0 g	1.05 (0.77–1.43)	0.746	1.17 (0.85–1.60)	0.344	1.24 (0.79–1.93)	0.352
>30.0 g of alcohol/day	1.42 (0.99–2.03)	0.054	**1.77 (1.07–2.92)**	**0.025**	1.35 (0.68–2.66)	0.389
Women						
Nondrinkers (ref.)	1.00		1.00		1.00	
0.1–5.0 g	**0.83 (0.72–0.95)**	**0.008**	1.01 (0.87–1.16)	0.938	0.91 (0.74–1.11)	0.350
5.1–10.0 g	**0.71 (0.58–0.87)**	**0.001**	1.00 (0.80–1.24)	0.969	0.99 (0.73–1.34)	0.959
10.1–15.0 g	**0.58 (0.40–0.83)**	**0.003**	0.91 (0.62–1.32)	0.614	1.07 (0.64–1.80)	0.786
>15.0 g of alcohol/day	**0.62 (0.44–0.89)**	**0.009**	0.97 (0.67–1.40)	0.857	0.87 (0.52–1.45)	0.595

Model I—unadjusted; model II—adjusted for age, marital status, education, smoking, coffee consumption and physical activity; model III—adjusted for age, marital status, education, smoking, coffee consumption, physical activity and BMI. The numbers in bold indicate statistically significant results. Ref.: Reference level.

**Table 4 nutrients-11-02764-t004:** Odds ratios and 95% confidence intervals for elevated blood pressure.

Elevated Blood Pressure or Antihypertensive Drug Treatment/Alcohol Consumption	Model I Unadjusted OR (95% CI)	*p*	Model II Adjusted OR (95% CI)	*p*	Model III Adjusted + BMI OR (95% CI)	*p*
Men						
Nondrinkers (ref.)	1.00		1.00		1.00	
0.1–10.0 g	0.95 (0.75–1.21)	0.674	0.90 (0.71–1.15)	0.409	0.90 (0.70–1.15)	0.396
10.1–20.0 g	1.25 (0.93–1.67)	0.138	1.28 (0.95–1.73)	0.104	1.25 (0.92–1.69)	0.152
20.1–30.0 g	1.07 (0.74–1.55)	0.706	1.15 (0.79–1.67)	0.464	1.11 (0.76–1.63)	0.577
>30.0 g of alcohol/day	**2.57 (1.53–4.30)**	**<0.001**	**2.76 (1.64–4.65)**	**<0.001**	**2.53 (1.49–4.30)**	**0.001**
Women						
Nondrinkers (ref.)	1.00		1.00		1.00	
0.1–5.0 g	**0.76 (0.67–0.87)**	**<0.001**	0.90 (0.78–1.03)	0.117	0.88 (0.77–1.02)	0.084
5.1–10.0 g	**0.68 (0.55–0.82)**	**<0.001**	0.92 (0.75–1.14)	0.450	0.95 (0.76–1.17)	0.614
10.1–15.0 g	0.72 (0.50–1.03)	0.072	1.09 (0.75–1.58)	0.665	1.17 (0.80–1.71)	0.424
>15.0 g of alcohol/day	0.79 (0.56–1.13)	0.204	1.11 (0.77–1.61)	0.569	1.09 (0.75–1.59)	0.650

Model I—unadjusted; model II—adjusted for age, marital status, education, smoking, coffee consumption and physical activity; model II—adjusted for age, marital status, education, smoking, coffee consumption, physical activity and BMI. The numbers in bold indicate statistically significant results. Ref.: Reference level.

**Table 5 nutrients-11-02764-t005:** Odds ratios and 95% confidence intervals for increased glucose concentration.

Increased Glucose Concentration or Drug Treatment of Elevated Glucose/Alcohol Consumption	Model I Unadjusted OR (95% CI)	*p*	Model II Adjusted OR (95% CI)	*p*	Model III Adjusted + BMI OR (95% CI)	*p*
Men						
Nondrinkers (ref.)	1.00		1.00		1.00	
0.1–10.0 g	0.99 (0.82–1.19)	0.919	1.00 (0.82–1.20)	0.964	1.00 (0.82–1.21)	0.971
10.1–20.0 g	1.15 (0.92–1.43)	0.225	1.22 (0.97–1.52)	0.087	1.19 (0.95–1.50)	0.133
20.1–30.0 g	1.14 (0.86–1.50)	0.372	1.26 (0.95–1.68)	0.113	1.25 (0.93–1.67)	0.135
>30.0 g of alcohol/day	**1.57 (1.16–2.14)**	**0.004**	**1.70 (1.24–2.32)**	**0.001**	**1.58 (1.15–2.18)**	**0.005**
Women						
Nondrinkers (ref.)	1.00		1.00		1.00	
0.1–5.0 g	0.88 (0.78–1.01)	0.061	1.04 (0.91–1.18)	0.597	1.04 (0.91–1.19)	0.569
5.1–10.0 g	1.17 (0.72–1.09)	0.242	1.16 (0.94–1.44)	0.159	1.24 (0.99–1.54)	0.056
10.1–15.0 g	1.17 (0.82–1.66)	0.398	**1.65 (1.14–2.38)**	**0.008**	**1.84 (1.26–2.70)**	**0.002**
>15.0 g of alcohol/day	**0.65 (0.44–0.96)**	**0.032**	0.85 (0.57–1.27)	0.436	0.86 (0.60–1.29)	0.466

Model I—unadjusted; model II—adjusted for age, marital status, education, smoking, coffee consumption and physical activity; model III—adjusted for age, marital status, education, smoking, coffee consumption, physical activity and BMI. The numbers in bold indicate statistically significant results. Ref.: Reference level.

**Table 6 nutrients-11-02764-t006:** Odds ratios and 95% confidence intervals for decreased HDL-cholesterol concentration.

Decreased HDL Concentration or Drug Treatment for Reduced HDL-Cholesterol/Alcohol Consumption	Model I Unadjusted OR (95% CI)	*p*	Model II Adjusted OR (95% CI)	*p*	Model III Adjusted + BMI OR (95% CI)	*p*
Men						
Nondrinkers (ref.)	1.00		1.00		1.00	
0.1–10.0 g	0.97 (0.79–1.19)	0.778	0.97 (0.79–1.20)	0.777	0.97 (0.78–1.20)	0.779
10.1–20.0 g	**0.71 (0.55–0.91)**	**0.007**	**0.73 (0.56–0.94)**	**0.016**	**0.70 (0.54–0.91)**	**0.008**
20.1–30.0 g	**0.57 (0.41–0.80)**	**0.001**	**0.61 (0.43–0.86)**	**0.005**	**0.60 (0.42–0.85)**	**0.004**
>30.0 g of alcohol/day	**0.51 (0.35–0.75)**	**0.001**	**0.51 (0.35–0.76)**	**0.001**	**0.47 (0.32–0.70)**	**<0.001**
Women						
Nondrinkers (ref.)	1.00		1.00		1.00	
0.1–5.0 g	**0.79 (0.70–0.89)**	**<0.001**	0.93 (0.82–1.06)	0.261	0.93 (0.82–1.05)	0.249
5.1–10.0 g	**0.60 (0.49–0.74)**	**<0.001**	**0.81 (0.65–0.99)**	**0.045**	0.83 (0.67–1.02)	0.083
10.1–15.0 g	**0.46 (0.31–0.69)**	**<0.001**	**0.66 (0.44–0.99)**	**0.048**	0.68 (0.44–1.03)	0.070
>15.0 g of alcohol/day	**0.41 (0.27–0.61)**	**<0.001**	**0.55 (0.36–0.82)**	**0.004**	**0.54 (0.35–0.82)**	**0.004**

Model I—unadjusted; model II—adjusted for age, marital status, education, smoking, coffee consumption and physical activity; model III—adjusted for age, marital status, education, smoking, coffee consumption, physical activity and BMI. The numbers in bold indicate statistically significant results; Ref.: Reference level.

**Table 7 nutrients-11-02764-t007:** Odds ratios and 95% confidence intervals for increased triglycerides concentration.

Increased Triglycerides Concentration or Drug Treatment for Elevated Triglycerides/Alcohol Consumption	Model I Unadjusted OR (95% CI)	*p*	Model II Adjusted OR (95% CI)	*p*	Model III Adjusted + BMI OR (95% CI)	*p*
Men						
Nondrinkers (ref.)	1.00		1.00		1.00	
0.1–10.0 g	1.03 (0.85–1.24)	0.794	1.04 (0.86–1.26)	0.703	1.04 (0.85–1.27)	0.687
10.1–20.0 g	1.07 (0.85–1.33)	0.580	1.06 (0.85–1.34)	0.594	1.03 (0.82–1.31)	0.775
20.1–30.0 g	0.99 (0.75–1.32)	0.965	1.02 (0.76–1.37)	0.880	1.01 (0.75–1.36)	0.959
>30.0g of alcohol/day	1.30 (0.95–1.77)	0.097	1.29 (0.95–1.77)	0.107	1.19 (0.87–1.64)	0.277
Women						
Nondrinkers (ref.)	1.00		1.00		1.00	
0.1–5.0 g	**0.85 (0.76–0.96)**	**0.011**	1.02 (0.90–1.16)	0.794	1.02 (0.89–1.16)	0.795
5.1–10.0 g	**0.74 (0.61–0.91)**	**0.003**	1.01 (0.82–1.24)	0.934	1.04 (0.84–1.28)	0.713
10.1–15.0 g	**0.62 (0.42–0.91)**	**0.014**	0.90 (0.61–1.33)	0.592	0.93 (0.62–1.39)	0.734
>15.0 g of alcohol/day	**0.55 (0.37–0.80)**	**0.002**	0.74 (0.50–1.09)	0.125	0.74 (0.50–1.10)	0.133

Model I—unadjusted; model II—adjusted for age, marital status, education, smoking, coffee consumption and physical activity; model III—adjusted for age, marital status, education, smoking, coffee consumption, physical activity and BMI. The numbers in bold indicate statistically significant results; Ref.: Reference level.

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
