# Peer review of "Consumption of Alcoholic Beverages and the Prevalence of Metabolic Syndrome and Its Components"

_nutrients, 2019, doi:10.3390/nu11112764_

Round 1
Reviewer 1 Report
Comments and suggestions to manuscript entitled “Consumption of alcoholic beverages and the prevalence of metabolic syndrome and its components” by Suliga et al. (reference Nutrients 616765)
In addition to my comments included in the file, please give any explanation to the absence of significant relationship between alcohol consumption and triglyceride levels.
The manuscript of Suliga et al. aims to analyze the relationship between the current consumption of alcoholic beverages and the prevalence of metabolic syndrome and its components in men and women. Although this topic has been already checked by some authors, controversial results are able on the relationship between alcohol consumption and chronic diseases. In addition, some factors considered in the metabolic syndrome diagnosis are affected by the alcohol consumption following a non-linear relationship. More, the positive effects attributed to some alcoholic beverage are mainly due to their minor compounds. Thus, any related paper can help to give light on this topic.
Major points
1.- Major strength of the paper is the high number of subjects tested while the imprecisely determining the amount and frequency of alcoholic beverages, together with the absence of dietary factors affecting metabolic syndrome factors (e.g. energy, carbohydrates, omega-3 PUFA consumption) are its major limitations.
2.-A hypothesis is missing. Please add one before aims in text and abstract.
1.- No information on drugs used for arresting metabolic syndrome components is given.
2.- Differences between alcohol consumption derived from wine and/or beer vs other alcoholic beverage are not shown.
3.- In formation on Physical activity (sedentary, moderate, medium, high levels) have been not considered.
Minor points
Please explain why you have selected the IDF classification and not other extensively used classification.
When defining Metabolic syndrome factors according to the International Diabetes Federation Task force, please add after each factor the following “....drug treatment or previously diagnosed of … (or similar).
Units for total physic activity expressed in metabolic equivalents are distinctly set in text (MET/min/week-1) and MTS/week in table.
Give clear information (references) on the method used for HDL-cholesterol and triglyceride determinations. I believe this was other limitation of the study as determination was performed in different labs and according to different analytical methods.
Support internal and external quality control data (%CV for each determination and give reference number of the controllers).
Table 1. Please distinct between parametric and non-parametric tests/data for the P values. Were single widow/widower and Married/in a relationship tested together by Squared Chi?
To improve reading, avoid several separating lines in this table. Just keep general ones. Thus, delete lines between Married and widow conditions; the same for different alcohol consumption degree and for different smoking conditions.
Widow and widower are wrongly written.
A graphical abstract will be welcome

Author Response
Rev 1
Major points
1.- Major strength of the paper is the high number of subjects tested while the imprecisely determining the amount and frequency of alcoholic beverages, together with the absence of dietary factors affecting metabolic syndrome factors (e.g. energy, carbohydrates, omega-3 PUFA consumption) are its major limitations.
Information on the lack of the data above, is included in limitations.
2.-A hypothesis is missing. Please add one before aims in text and abstract.
As suggested by the reviewer, a hypothesis was added to the summary and the text.
1.- No information on drugs used for arresting metabolic syndrome components is given.
Information on drugs used to treat metabolic syndrome components has been added to the text.
2.- Differences between alcohol consumption derived from wine and/or beer vs other alcoholic beverage are not shown.
In the study group over 81% of men and 65% of women declared drinking more than one type of alcohol (beer + wine or vodka + wine, e.t.c.). That is why we decided to carry out the alcohol analysis together.
3.- In formation on Physical activity (sedentary, moderate, medium, high levels) have been not considered.
Physical activity was analyzed together because sitting time and walking did not show a significant relationship with the variables studied but were significantly correlated with each other. Therefore, collinearity diagnostics did not allow for the analysis of all three types of physical activity.
In addition to my comments included in the file, please give any explanation to the absence of significant relationship between alcohol consumption and triglyceride levels.
The discussion has been supplemented according to the reviewer's instructions.
Minor points
Please explain why you have selected the IDF classification and not other extensively used classification.
The explanation has been included in the article.
When defining Metabolic syndrome factors according to the International Diabetes Federation Task force, please add after each factor the following “....drug treatment or previously diagnosed of … (or similar).
The definition has been supplemented according to the reviewer’s suggestions
Units for total physic activity expressed in metabolic equivalents are distinctly set in text (MET/min/week-1) and MTS/week in table.
Units for total physical activity have been corrected
Give clear information (references) on the method used for HDL-cholesterol and triglyceride determinations. I believe this was other limitation of the study as determination was performed in different labs and according to different analytical methods.
The above information has been added. However, we would like to clarify that biochemical analyzes were carried out in one laboratory using the same method. Only blood samples for testing had been taken at various locations.
Support internal and external quality control data (%CV for each determination and give reference number of the controllers).
The above information has been added.
Table 1. Please distinct between parametric and non-parametric tests/data for the P values. Were single widow/widower and Married/in a relationship tested together by Squared Chi?
The statistical tests that had been carried out are included in Table 1
To improve reading, avoid several separating lines in this table. Just keep general ones. Thus, delete lines between Married and widow conditions; the same for different alcohol consumption degree and for different smoking conditions.
The table has been corrected as suggested by the reviewer
Widow and widower are wrongly written.
It has been corrected
A graphical abstract will be welcome
The time to prepare an improved version of the study under "Nutrients" is unfortunately very short, so we were not able to develop a graphical abstract.

Reviewer 2 Report
Suliga et al investigated the association of alcohol consumption with the prevalence of metabolic syndrome. Even though similar studies have been done in other cohorts, the current study used the PONS cohort and provided another piece of information. The manuscript was well written and the results were nicely presented. I only have two minor comments:
The titles of Tables 2-7 should be more succinct and summarized. The current tiles can be put as legends. The discussion mainly focused on the description of other studies and comparing them with the current study. More discussion of the potential reasons of the different observations between men and women is suggested.Author Response
Rev 2
Suliga et al investigated the association of alcohol consumption with the prevalence of metabolic syndrome. Even though similar studies have been done in other cohorts, the current study used the PONS cohort and provided another piece of information. The manuscript was well written and the results were nicely presented. I only have two minor comments:
The titles of Tables 2-7 should be more succinct and summarized. The current tiles can be put as legends.
Table titles and legends have been corrected as suggested by the reviewer
The discussion mainly focused on the description of other studies and comparing them with the current study. More discussion of the potential reasons of the different observations between men and women is suggested.
The discussion has been expanded.

Round 2
Reviewer 1 Report
Although in the first version it was stated that association was significant in the second one it was deleted. This is true than normally we say that not association (or correlation) was found when this was not significant but this is incorrect. As allways there are association except in case of value 0. Please maintain significant or not significant association instead of associated or not associated